# Lymphoproliferations in People Living with HIV: Oncogenic Pathways, Diagnostic Challenges, and New Therapeutic Opportunities

**DOI:** 10.3390/cancers17132088

**Published:** 2025-06-22

**Authors:** Riccardo Dolcetti, Emanuela Vaccher, Antonino Carbone

**Affiliations:** 1Peter MacCallum Cancer Centre, Melbourne, VIC 3000, Australia; 2Sir Peter MacCallum Department of Oncology, The University of Melbourne, Melbourne, VIC 3000, Australia; 3Department of Microbiology and Immunology, The University of Melbourne, Melbourne, VIC 3000, Australia; 4Centro di Riferimento Oncologico, Istituto di Ricovero e Cura a Carattere Scientifico, National Cancer Institute, 33081 Aviano, Italy; evaccher@cro.it

**Keywords:** human immunodeficiency virus, lymphomas, Epstein–Barr virus, Kaposi’s sarcoma-associated herpesvirus, lymphomagenesis, combined antiretroviral therapy, immunotherapy

## Abstract

People living with Human Immunodeficiency Virus (HIV) are at significantly increased risk of developing lymphoproliferative disorders. Although the advent of combination antiretroviral therapy (cART) has markedly improved survival, lymphomas remain a major cause of morbidity and mortality in this population. In this review, we discuss the key features of this heterogeneous group of tumors, highlight current diagnostic and therapeutic challenges, and explore emerging immunotherapeutic strategies aimed at improving lymphoma outcomes in individuals with HIV.

## 1. Introduction

With the widespread implementation of combination antiretroviral therapy (cART), lymphoma has become the most common malignancy among people living with HIV (PLWH) in many developed countries [1,2,3,4]. Over the last two decades, the incidence of certain subtypes, such as diffuse large B cell lymphoma (DLBCL) and primary central nervous system lymphomas (PCNSLs), has decreased, while the incidence of Burkitt lymphoma (BL) has remained stable, and Hodgkin lymphoma (HL) has increased [1]. These lymphomas are generally characterized by high-grade features, frequent extranodal involvement, advanced-stage presentation, and marked histopathologic heterogeneity, often posing significant diagnostic challenges [1]. The 5th edition of the World Health Organization (WHO) Classification of Haematolymphoid Tumors now includes a dedicated category for HIV-associated lymphomas [5]. In this review, we provide an overview of the current understanding of the complex pathogenesis of lymphoproliferation in PLWH, outline their main clinicopathologic features, discuss current therapeutic strategies, and explore emerging opportunities for immunotherapy aimed at improving outcomes in this population.

## 2. Pathogenesis of HIV-Associated Lymphoproliferative Disorders

The pathogenesis of lymphomas in PLWH is multifaceted and complex, sharing several features with lymphoproliferative disorders associated with other congenital or acquired immunodeficiencies [6]. Indirect factors related to HIV-induced immunosuppression, such as chronic antigenic stimulation, dysregulated cytokine production, and co-infection with oncogenic viruses, are recognized contributors to lymphomagenesis in this setting [7]. However, emerging evidence challenges the traditional view, suggesting that HIV itself may also play a more direct role in promoting lymphoma development [8].

### 2.1. Indirect Mechanisms

The pathogenesis of HIV-associated lymphoproliferative disorders involves multiple factors, primarily, though not exclusively, linked to the immunosuppressive effects of HIV infection.

HIV-induced immunosuppression and chronic antigenic stimulation. HIV-induced depletion of CD4^+^ T cells leads to a progressive decline in immune competence, impairing the host’s ability to control oncogenic infections and pre-malignant lymphocyte clones [9]. This immune dysfunction facilitates the unchecked expansion of lymphocytes harboring predisposing genetic alterations or oncogenic virus infections, thereby increasing lymphoma risk. Despite the virologic control achieved with cART, T cell dysfunction persists. Specifically, the sustained expression of inhibitory checkpoint receptors, such as PD-1, TIM-3, or TIGIT, on T cells limits their cytotoxic potential against emerging tumor cells [10]. Furthermore, chronic immune activation and inflammation—hallmarks of HIV infection—also contribute to persistent antigenic stimulation, which can drive lymphomagenesis [7,8]. HIV infection also disrupts cytokine homeostasis, promoting the overproduction of pro-inflammatory mediators such as IL-6, which support the survival and proliferation of neoplastic lymphocytes [11]. Macrophages are particularly affected: HIV infection impairs their function and contributes to both viral persistence and tumor-promoting [12]. For instance, the HIV viral protein U (Vpu) degrades the bone marrow stromal antigen 2 (BST2), a macrophage protein that restricts viral replication, thereby facilitating chronic inflammation and persistent infection [13,14]. Moreover, HIV-infected macrophages upregulate several microRNAs, including miRNA-99 and miRNA-146a, which stimulate the release of pro-inflammatory cytokines (e.g., TNFα, IL-6, and IL-1β), and dampen innate immune responses [15,16]. HIV-encoded Tat and Nef proteins further impair immunity by inducing Indoleamine 2,3-dioxygenase (IDO), leading to Th17 cell depletion, regulatory T cell expansion, and reduced CD8^+^ T cell proliferation—all of which foster immune escape and tumor development [17,18,19]. Natural killer (NK) cells are also affected. HIV infection impairs the activation of CD56^bright^ NK cells and downregulates critical activating receptors (e.g., NCRs and NKG2D), largely due to chronic exposure to their ligands [20,21]. A dysfunctional CD56^−^CD16^+^ NK cell subset expands at the expense of the cytotoxic CD56^dim^CD16^+^ population, with reduced expression of NKp30 and NKp46 [22,23]. These altered NK cells exhibit impaired cytokine production, defective crosstalk with other immune cells, and diminished cytotoxicity toward tumor targets [22,23]. Although cART partially restores NK cell function [23], residual abnormalities persist and may contribute to ongoing lymphoma risk in PLWH.

Oncogenic virus co-infection. The role of co-infecting viruses in the development of HIV-associated malignancies has been extensively investigated, with Epstein–Barr virus (EBV) and Kaposi Sarcoma Herpesvirus (KSHV) being strongly implicated in specific HIV-related lymphoma subtypes [24,25,26,27]. In both cases, HIV-induced immunosuppression undermines immune surveillance, enabling the unrestricted proliferation of virus-infected cells and the aberrant expression of oncogenic viral proteins that directly drive lymphomagenesis. Notably, the degree of HIV-induced immunosuppression correlates with the type of EBV- or KSHV-associated lymphoma that arises in PLWH [24,25,26,27]. Before the widespread use of cART, severe immunodeficiency led to a high incidence of aggressive EBV^+^ DLBCLs and immunoblastic lymphomas. These neoplasms typically exhibit the EBV latency III pattern, characterized by the expression of the full repertoire of EBV nuclear and membrane proteins [28]. The advent of cART significantly reduced the incidence of such lymphomas [1], reflecting improved immune control of EBV. In contrast, EBV-associated lymphomas arising in PLWHs with relatively preserved immune function—such as BL and HL—demonstrate more restricted patterns of EBV gene expression. BL is associated with EBV latency I (expression of EBNA1 only), while HL is linked to latency II (expression of EBNA1, LMP1, and LMP2) [25,26]. The incidence of these lymphoma types has not significantly changed following cART introduction [1], suggesting that their pathogenesis may be influenced by mechanisms beyond HIV-induced immunosuppression. Of note, both BL and HL originate in germinal centers—an anatomical niche where HIV replication can persist despite cART [1,5]. Within this microenvironment, the expression of EBV nuclear antigen 6 (EBNA6) by EBV-infected B cells may induce activation-induced cytidine deaminase (AICDA), a DNA-editing enzyme that can drive 2oncogogenic mutations [29,30]. The synergistic oncogenic potential of HIV and EBV has been demonstrated by elegant studies in humanized mouse models, where co-infection enhances systemic EBV replication, promotes immune activation, and facilitates EBV-driven lymphomagenesis [31]. Beyond its latent and lytic repertoire, EBV also contributes to lymphomagenesis via the expression of viral microRNAs. These miRNAs are encoded in two genomic regions—BamHI-A region rightward transcript (BART) and BamHI-H rightward fragment 1 (BHRF1)—and modulate host pathways involved in apoptosis, proliferation, immune evasion, and the tumor microenvironment [32,33]. BART miRNAs are broadly expressed across all latency types, particularly latency I and II, while BHRF1 miRNAs are largely restricted to latency III [32,33].

In addition to Kaposi’s sarcoma, the oncogenic γ-herpesvirus KSHV is etiologically linked to certain B cell malignancies, notably primary effusion lymphoma (PEL) [34], and a subset of multicentric Castleman disease (MCD) [35]. KSHV appears to persist preferentially in immunoglobulin lambda-expressing B cells, which are enriched in both PEL and MCD tissues [36,37], although the precise reservoir of KSHV in asymptomatic carriers remains poorly defined. KSHV-driven lymphomagenesis is primarily attributed to latent infection. Most KSHV-infected tumor cells express a limited subset of latent viral proteins, with Latency-Associated Nuclear Antigen (LANA) being the most consistently expressed [24]. Functionally analogous to EBV’s EBNA-1, LANA is essential for maintaining the viral episome during latency and contributes to oncogenesis by modulating both viral and host gene expression through interactions with cellular transcription factors [24]. Other latent KSHV proteins—such as viral FLICE inhibitory protein (vFLIP), v-cyclin, Kaposin, and viral interferon regulatory factor 3 (vIRF-3)—may also be expressed in PEL and MCD, though typically at lower frequencies [24]. KSHV lytic proteins also play roles in lymphomagenesis by disrupting apoptotic pathways, promoting angiogenesis, remodeling the tumor microenvironment, and enabling immune evasion. Among these, viral interleukin 6 (vIL-6) is particularly relevant, especially in MCD. vIL-6 promotes vascular endothelial growth factor (VEGF)-mediated vascular permeability, drives IL-6-dependent inflammation, and directly supports the growth of PEL cells [38,39,40]. Importantly, the vast majority of PEL cases are also co-infected with EBV, and there is growing evidence of a pathogenic interplay between the two viruses. KSHV has been shown to trigger lytic EBV replication, which appears to contribute to lymphomagenesis [24]. Notably, studies in humanized mouse models demonstrated that co-infection with KSHV and an EBV strain incapable of lytic replication led to a lower incidence of lymphoma, suggesting that abortive lytic EBV reactivation—rather than full viral replication—may be critical in promoting tumor development [41]. Complete EBV lytic replication likely reduces lymphoma formation due to cell lysis and subsequent clearance of infected cells.

Contribution of cART. While cART significantly improves overall immune function and reduces HIV-related mortality, its long-term use—particularly in individuals with advanced HIV infection—only partially reverses the metabolic alterations, chronic inflammation, and immune dysregulation that can indirectly contribute to the development of lymphoproliferative disorders [42,43]. In some cases, rapid immune restoration following cART initiation may lead to an exaggerated inflammatory response against latent viral infections, a condition known as Immune Reconstitution Inflammatory Syndrome (IRIS) [44]. IRIS has been associated with the emergence or unmasking of lymphoproliferative processes, although its precise role in HIV-related lymphomagenesis remains unclear. Further studies are needed to delineate the direct and indirect contribution of cART to lymphoma development in PLWH.

Microenvironmental factors. The tumor microenvironment (TME) of HIV-associated lymphomas is highly complex and plays a pivotal role in disease development by fostering immunosuppression, facilitating tumor immune evasion, and supporting tumor progression [7,45]. HIV-associated lymphomas often arise and evolve within a dynamic and interactive microenvironment, where malignant lymphocytes coexist and communicate with various immune and stromal components. This interaction mirrors the physiological relationship between normal B cells and their microenvironment [7,45]. In most cases, the lymphoma’s growth remains partially dependent on signals from the TME. However, this interdependence may be lost when genetic alterations endow tumor cells with full autonomy. A prototypical example is BL, in which the t(8;14) chromosomal translocation leads to constitutive activation of the c-MYC oncogene, driving uncontrolled proliferation independent of microenvironmental cues [46].

### 2.2. Direct Mechanisms

Despite the partial restoration of immune function achieved with cART, PLWH continues to exhibit a disproportionately high incidence of lymphomas, including histological subtypes with distinctive cyto-histopathological features [1]. Emerging evidence suggests that, beyond immune suppression, HIV may also directly contribute to lymphomagenesis through mechanisms involving the biological activity of viral proteins [5]. Although HIV does not infect B lymphocytes—the primary target cells in most HIV-associated lymphomas—it can release viral proteins into the surrounding microenvironment. HIV-encoded proteins, such as gp120, Nef, Tat, and HIV matrix protein p17 (p17), can interact with or be internalized by bystander cells, including B lymphocytes, where they activate intracellular signaling pathways. These interactions may promote chronic antigenic stimulation, dysregulation of cellular processes, and ultimately contribute directly to the initiation and progression of lymphomagenesis [8,47,48].

Following its release in lymphoid tissues, the HIV Tat protein can functionally interact with and enter B lymphocytes, where it influences key cellular processes, including cell cycle regulation, apoptosis, redox homeostasis, and gene expression [43]. Tat has been shown to upregulate the production of pro-survival and proliferative cytokines such as IL-6 and IL-10, thereby enhancing B cell activation and proliferation [49,50]. Additionally, Tat induces the expression of the DNA methyltransferase 1 enzyme, implicating it in the epigenetic dysregulation of host gene expression and suggesting a potential role in lymphomagenesis [51]. Early cellular responses to Tat include oxidative stress, DNA damage, and increased genomic instability in B cells [52]. Tat also activates the Akt/mTORC1 signaling pathway and the expression of AICDA by suppressing its transcriptional repressors [53,54,55], further contributing to mutagenesis and chromosomal instability. More recently, chronic Tat expression has been shown to downregulate HLA-DR expression via the inhibition of the NF-κB pathway, impairing CD4^+^ T cell-mediated immune recognition of EBV-transformed B cells and thus facilitating immune evasion [56]. In vivo studies have confirmed the oncogenic potential of Tat, with transgenic mice expressing Tat developing a range of different neoplasms, including lymphomas [57]. In addition to Tat, the HIV-encoded Nef protein has been implicated in lymphomagenesis by promoting AICDA and c-MYC expression, both of which are central to genomic instability and the formation of pathogenic chromosomal translocations [58].

Multiple lines of evidence support a potential lymphomagenic role for the HIV matrix protein p17. HIV-infected cells can release virion-free p17 independently of active viral protease activity, and this protein has been detected in the plasma and lymphoid tissues of PLWH [59,60]. Notably, p17 persists in lymphoid tissues even during cART and in the absence of active HIV replication [60]. This long-term presence within the tissue microenvironment suggests that p17 may serve as a chronic stimulus for B cell activation and proliferation. Supporting this hypothesis, the expression of p17 in mice transgenic for a defective HIV-1 provirus has been shown to result in lymphoma development [61]. Furthermore, specific p17 variants (vp17)—derived either from a Ugandan HIV-1 A1 strain or from individuals with HIV-associated non-Hodgkin lymphoma—demonstrate distinct biological activity compared to prototype clade B p17 or p17 proteins isolated from PLWH without lymphoma [62,63]. These vp17s exhibit unique amino acid insertions in their C-terminal region, resulting in marked conformational changes that expose a functional epitope in the N-terminal domain of the protein [62,63,64]. Recent studies have shown that this functional epitope interacts with the Protease-Activated Receptor 1, triggering downstream epithelial growth factor receptor (EGFR) transactivation and activation of the phosphoinositide 3-kinase (PI3K)-Akt signaling pathway—ultimately promoting B cell proliferation and clonogenicity [65]. Consistent with these mechanistic insights, retrospective cohort analyses have demonstrated a significantly higher prevalence of vp17s in PLWH with lymphoma compared to those without lymphoma [66].

The inflammatory microenvironment and the chronic B cell stimulation induced by HIV-encoded proteins and other exogenous factors may promote polyclonal or oligoclonal expansion of aberrant B lymphocytes. This proliferation is sustained by the dysregulated production of B cell growth-promoting cytokines, such as IL-6 and IL-10. In this context of persistent immune activation, the risk increases for monoclonal B cell expansions to acquire genetic lesions critical for lymphomagenesis. Indeed, recent studies have identified unique B cell subsets in PLWH that display pre-malignant features and may represent early stages in the evolution toward overt lymphoma [67]. Notably, HIV virions released from infected CD4^+^ T cells can incorporate the costimulatory molecule CD40 ligand, thereby gaining the capacity to activate B lymphocytes in a manner that mimics physiological T–B interactions [68,69]. Regardless of the initiating stimulus, chronic B cell activation is closely associated with the induction of AICDA, a key enzyme in somatic hypermutation and class–switch recombination [70]. Elevated AICDA expression has been detected in circulating B lymphocytes of HIV^+^ individuals prior to lymphoma diagnosis, particularly in those who later developed BL, a disease strongly associated with immunoglobulin gene translocations [70]. Furthermore, HIV-related dysregulation of B cell receptor (BCR) signaling has been proposed as an additional mechanism contributing to lymphomagenesis. It has been hypothesized that HIV may remodel the structure of the BCR through AICDA and recombinant-activating gene upregulation while simultaneously acting as a chronic antigen that activates downstream signaling pathways, including PI3K and MAPK, partly through reduced expression of the inhibitory receptor CD300a [71].

## 3. Pathologic Features of HIV-Associated Lymphoproliferations

Polymorphic lymphoproliferative disorders, which span a spectrum of malignant potential, are morphologically and clinically heterogeneous. They disrupt the normal tissue architecture and are frequently associated with necrosis [72,73,74]. Although classically associated with post-transplant immunosuppression, polymorphic lymphoproliferative disorders have also been documented in PLWH [74,75,76,77,78,79,80]. Histologically, these lesions are characterized morphologically by a heterogeneous, or polymorphic, cell population representing various stages of B cell differentiation—typically expressing CD20 and/or PAX5 and generally of non-germinal center origin—alongside plasma cells, other inflammatory cells, and Reed–Sternberg-like cells [81]. A variable proportion of the Reed–Sternberg-like cells are EBV^+^ and express the EBV oncogenic latent membrane protein-1 (LMP1) [71,72,74,75,78,79,80,81,82,83,84,85,86,87]. These lesions typically contain monoclonal B cell populations and oligoclonal T cell infiltrates, the latter likely representing a reactive response to the expansion of EBV-infected B cells [70,71,74,75,76,77,79,81,82,83,88].

HIV-associated malignant lymphomas are classified using the same morphologic, immunophenotypic, and genotypic criteria applied to lymphomas in immunocompetent individuals [5,88,89] and, similarly, are broadly categorized into Hodgkin lymphomas (HLs) and non-Hodgkin lymphomas (NHLs). Compared to their counterparts in immunocompetent patients, HIV-associated lymphomas are more frequently EBV-positive. Moreover, certain lymphoma subtypes show a predilection for PLWH, including plasmablastic lymphoma (PBL) and primary effusion lymphoma (PEL), along with its solid variant, extra-cavitary PEL (EC-PEL) [89,90] (Table 1).

### 3.1. Diffuse Large B Cell Lymphoma (DLBCL)

HIV-associated DLBCL is the most common subtype of HIV-associated lymphoma [1,6]. While HIV-associated DLBCLs share many histopathological and molecular features with their counterparts in immunocompetent individuals, they differ in their higher frequency of EBV positivity and distinct biological characteristics. These lymphomas can originate from either germinal center B cells (GCB subtype) or non-germinal center activated B cells (ABC subtype) [1,6]. The ABC-like subtype is frequently EBV-positive, while the GCB-like subtype is typically EBV-negative [93,94,95]. Both subtypes exhibit increased vascular density compared to DLBCL in HIV-negative individuals, reflecting an altered tumor microenvironment [96]. Cytogenetically, the prevalence of MYC and BCL6 rearrangements in HIV-associated DLBCL (14.9% and 27.7%, respectively) is comparable to that in HIV-negative patients, whereas BCL2 rearrangements are less frequently observed (4.3%) [97]. The TME in HIV-associated DLBCL includes a prominent population of tumor-associated macrophages (TAMs), which have been implicated in tumor progression [98]. These TAMs are often skewed toward an M2-like, pro-tumoral phenotype under the influence of cytokines such as IL-4 and IL-10 [99]. In addition to supporting tumor growth, M2-like macrophages promote angiogenesis in the tumor milieu of HIV-associated DLBCLs [7,45]. Importantly, both tumor cells and infiltrating macrophages in aggressive HIV-associated DLBCLs have been shown to express the checkpoint molecule programmed cell death ligand 1 (PD-L1) [100,101], highlighting the potential utility of immune checkpoint inhibitors targeting the PD-1/PD-L1 axis in this context.

### 3.2. Classic Hodgkin Lymphoma (HL) in PLWH

In high-income countries, PLWH have a 5- to 26-fold increased risk of developing HL compared to the general population [4]. The highest risk period for HL diagnosis often coincides with the months following initiation of cART, during which the inflammatory IRIS may occur [102,103]. Unlike NHL, the incidence of HL in PLWH has remained relatively stable or has declined only modestly in the cART era [3,4]. Notably, PLWH on effective cART—with suppressed viral load and restored CD4^+^ T cell counts—still exhibit a ninefold higher risk of HL than HIV-negative individuals [104], further implicating IRIS and immune dysregulation as potential contributors to HIV-associated HL pathogenesis [103,105]. Classic Hodgkin lymphoma (cHL) in the context of HIV is EBV^+^ in over 90% of cases [1,5,26,106,107,108] with frequent strong expression of the LMP-1 oncogenic EBV protein [109] (Figure 1). Although the immunophenotype of the neoplastic Hodgkin–Reed–Sternberg (HRS) cells closely resemble that observed in immunocompetent patients, the TME in PLWH is distinct, being characterized by a lower CD4 to CD8 T cell ratio and altered immune cell composition [26].

At the time of diagnosis, HIV-associated cHL typically presents with a relatively high median CD4^+^ T cell count is relatively high—ranging from approximately 275 to 306 cells/μL—indicating only moderate immunosuppression [109]. However, the number of functional and mature NK cells within HIV-associated HL tissues is reduced compared to HIV-negative HL cases, suggesting an impaired innate immune response [110]. The TME of cHL is complex and composed of various cellular and structural components, including B and T cells, fibroblasts, stromal cells, macrophages, mast cells, granulocytes, and an ECM [111,112]. The survival and proliferation of HRS cells are highly dependent on this supportive TME, particularly the presence of CD4^high^ T cells forming “rosettes” around HRS cells, which deliver essential growth-promoting signals [113].

Compared to HIV-unrelated EBV^+^ cases, the TME of HIV^+^EBV^+^ cHL is characterized by significantly lower proportions of CD8^+^ T cells co-expressing the inhibitory checkpoint molecules PD-1 and TIGIT. This reduction may reflect increased turnover, impaired T cell receptor (TCR) activation, or dysregulated PD-1/TIGIT expression. Notably, elevated levels of TIGIT ligands—CD155 and nectin 3—within the TME suggest that ligand/receptor interactions may contribute to the downregulation of TIGIT on CD8^+^ T cells [114]. Recent transcriptomic analyses have shown that relative to HIV-unrelated cHL, the HIV^+^EBV^+^ cHL TME exhibits a reduced expression of genes involved in TCR signaling and T cell stemness. Concurrently, these cases display increased activation of Gα and G-protein-coupled receptor signaling pathways, which are known to suppress T cell proliferation and inhibit IL-2 production—key mechanisms of T cell activation [115]. Additionally, a significant upregulation of ECM remodeling pathways was observed in HIV^+^EBV^+^ cHL, potentially interfering with effective adaptive immune responses. These findings support the notion that HIV infection may impair anti-tumor immunity in cHL through both direct effects on TCR signaling and indirect reorganization of the ECM. EBV infection further contributes to immune evasion by promoting the accumulation of CD68^high^ and CD163^high^ macrophages, consistent with an immunosuppressive M2-like phenotype. This is likely mediated in part by the EBV-encoded LMP1, which modulates the TME to favor tumor immune escape [116].

### 3.3. Epstein–Barr Virus-Positive Marginal Zone Lymphoma (EBV-MZL)

EBV-positive marginal zone lymphoma (EBV-MZL) has been reported in transplant recipients and in other settings of immunodeficiency or immune dysregulation, including PLWH [73,116,117,118]. Similar to MZL in immunocompetent patients, EBV-MZL in PLWH can arise at various anatomical sites—such as the skin, lung, salivary gland, stomach, and lymph nodes—and typically displays comparable morphologic features [116,117,118].

### 3.4. HIV-Associated Burkitt Lymphoma (BL)

BL accounts for approximately 25–40% of lymphomas occurring in PLWH [1,2,3]. Despite the introduction of cART, the incidence of HIV-associated BL has not declined [1,2,3,4]. PLWH with moderate immunodeficiency carry a lifelong risk of 10–20% for developing BL, a risk that is more strongly associated with cumulative HIV viremia than with recent immune status [3,4]. Clinically, HIV-associated BL presents with distinct features compared to BL in HIV-negative individuals [119,120]. Patients often present with B symptoms, advanced-stage disease, and frequent extranodal involvement—including bone marrow, oral cavity, gastrointestinal tract, and central nervous system [1]. Notably, HIV-associated BLs occurring in the context of relatively preserved CD4^+^ T cell counts are more likely to exhibit extranodal disease, B symptoms, frailty, and elevated lactate dehydrogenase levels [120]. EBV is detected in approximately 60% of HIV-associated BL cases, compared to about 20% in HIV-negative BL [121]. Histological features and EBV immunostaining are illustrated in Figure 2. Nearly all HIV-associated BLs harbor *MYC* gene rearrangements on chromosome 8. Importantly, the HIV Tat protein has been detected within BL tumor cells, where it enhances c-MYC expression through direct interaction with AP-1 sites on the c-MYC promoter [122]. In agreement with what we described above, these findings support a role for Tat in promoting oncogenic signaling within the BL TME, potentially contributing to the aggressiveness of this HIV-related lymphoma.

### 3.5. Other Rare HIV-Associated Lymphomas

Primary central nervous system lymphoma (PCNSL) is a rare but highly aggressive malignancy. In PLWH, the incidence of PCNSL is significantly increased, accounting for approximately 12–15% of HIV-associated lymphomas [123]. The vast majority of PCNSL cases (90–95%) are DLBCL, with a minority comprising BL or T-cell lymphomas. EBV infection is detected in 80–100% of PCNSL cases [123].

Plasmablastic lymphoma (PBL) and primary effusion lymphoma (PEL) are also rare and aggressive entities predominantly observed in PLWH [1,124,125]. While the exact incidence of HIV-associated PBL remains unclear, it accounts for approximately 2% of HIV-related lymphomas diagnosed in high-income countries during the cART era [1]. In contrast, in low- and middle-income countries (LMICs) with a high burden of HIV infection—such as South Africa—the incidence of PBL may be as high as 20% despite widespread cART availability [126]. However, high-quality epidemiological data on hematologic malignancies among PLWH in LMICs remain limited [4]. PBLs are consistently CD20-negative and EBV^+^ in about 75% of cases. It shows a marked male predominance and presents extranodally in over 95% of cases, with common sites including the oral cavity, gastrointestinal tract, skin, and bone [1]. The molecular pathogenesis of PBL involves dysregulation of *MYC* and activation of key oncogenic pathways such as JAK-STAT, RAS-RAF, and NOTCH, highlighting potential therapeutic targets for future investigation [124]. Despite this, clinical and biological insights into PBL remain limited, primarily based on case reports, underscoring the need for further research.

KSHV/HHV8 is an oncogenic virus linked to several HIV-associated malignancies: Kaposi sarcoma, NHL (PEL and DLBCL), MCD, and the Kaposi sarcoma inflammatory cytokine syndrome (KICS), and is classified as a human oncogenic virus [127,128]. The fifth edition of the WHO classification groups KSHV/HHV8-associated lymphoproliferative disorders into a single disease category, while KSHV/HHV8-associated MCD is classified under tumor-like lesions with B-cell predominance and immune dysregulation [5]. There are two clinical presentations of PEL: classic PEL, which involves serous effusions without tumor masses, and extra-cavitary PEL (EC-PEL), which manifests as nodal or extranodal masses with or without effusions. PELs are strongly associated with systemic inflammatory symptoms and frequently co-occur with other KSHV-associated diseases in over 80% of cases. These lymphomas most commonly arise in PLWH but can also be seen in post-transplant recipients and elderly patients with immune senescence [127,128]. While PELs in the HIV-infected patient population can be either EBV-positive or EBV-negative, cases in the elderly tend to be EBV-negative [92,129] (Figure 3).

There is a causal relationship between KSHV/Human Herpesvirus 8 (HHV8) and PEL, and this infection is a diagnostic requirement [128,129]. Another KSHV/HHV8-associated disorder, plasmablastic MCD, is characterized by polyclonal or oligoclonal B cells with unmutated immunoglobulin genes despite showing lambda light chain restriction. The disease typically arises in the context of immune dysregulation, and over 90% of cases occur in PLWH [130]. Given the shared pathogenesis, it is not uncommon for KSHV/HHV8-associated diseases to co-exist. PEL—often co-infected with EBV—along with PBL and KSHV/HHV8-positive DLBCL, have all been described in patients with MCD. These malignancies are generally associated with poor prognosis and often require aggressive systemic chemotherapy [131,132].

T-cell lymphomas are rare among PLWH, accounting for less than 5–10% of HIV-associated lymphomas. The most commonly reported subtypes include peripheral T-cell lymphoma, not otherwise specified [133], anaplastic large-cell lymphoma [134], and angioimmunoblastic T-cell lymphoma [135]. These lymphomas typically exhibit high-grade histologic features, a high proliferation index (Ki-67^+^), and expression of markers such as CD30 and CD4. Prognosis is generally poor, with significantly shorter median survival compared to HIV-negative individuals with similar lymphoma subtypes [136]. A distinct T cell entity, adult T-cell leukemia/lymphoma (ATLL), is etiologically linked to Human T-cell Lymphotropic Virus type 1 (HTLV-1) infection. Co-infection with HIV, particularly in HTLV-1-endemic regions such as Japan, the Caribbean, and sub-Saharan Africa, may accelerate disease progression, although only about 5% of HTLV-1 carriers develop ATLL [137]. EBV has also been implicated in the pathogenesis of certain HIV-associated T-cell lymphomas, especially in extranodal NK/T-cell lymphoma, nasal type. This lymphoma subtype, which is strongly associated with EBV, is more prevalent in distinct geographic regions, including parts of Asia and Latin America [138].

## 4. Treatment Strategies

Modern cART has significantly improved life expectancy and cancer treatment outcomes in PLWH by ensuring sustained HIV suppression and partial recovery of immune function [130,131,132]. For DLBCL, BL, and cHL, survival rates in PLWH now closely resemble those of HIV-negative patients when treated with full-dose chemotherapy in conjunction with cART. However, the prognosis remains poor for PBL and KSHV-associated lymphomas [1,2,139,140]. Prognostic outcomes in HIV-associated lymphomas are more strongly influenced by lymphoma-specific factors—such as those captured by the age-adjusted International Prognostic Index (IPI), BL- or PEL-specific IPI scores, and failure to achieve complete response (CR) to therapy—rather than by HIV-specific factors or EBV-related biomarkers [1,141,142,143,144]. Despite advancements, population-based studies have shown that disparities in overall and cancer-specific survival persist between PLWH and the general cancer population, particularly among socially disadvantaged or medically underserved groups [142,145,146,147,148,149]. Importantly, HIV infection should not be an exclusion criterion for enrollment in cancer clinical trials, and HIV-associated malignancies should be treated according to established standards of care [1,150,151,152].

The management of these malignancies requires a multidisciplinary approach, with concurrent administration of cART and antineoplastic therapy being critical for immune restoration and long-term disease control [1,152,153,154]. Given the potential for significant drug–drug interactions (DDIs), careful evaluation of interactions between antineoplastic agents, cART, and supportive medications is essential. Ritonavir- and cobicistat-boosted regimens are associated with numerous DDIs that may compromise treatment efficacy or tolerability. In contrast, integrase strand transfer inhibitors such as raltegravir, dolutegravir, and bictegravir offer more favorable DDI profiles and lead to faster reductions in HIV viremia [155,156]. Supportive care, including prophylaxis against opportunistic infections and the use of hematopoietic growth factors, is particularly important in patients with low CD4 cell count [1,150,152]. Furthermore, the growing burden of comorbidities reduced comorbidity-free survival in aging PLWH [130,157], necessitating vigilant monitoring and coordinated, multidisciplinary care strategies [1,158]. Post-treatment cancer surveillance programs should also be implemented, as PLWH are at increased risk for secondary malignancies, including Kaposi sarcoma, Human Papilloma Virus-associated anogenital and head and neck cancers, and additional NHLs [159,160,161,162,163].

### 4.1. Current Treatment Strategies for HIV-Related NHL

Recent immunodeficiency and cumulative HIV viremia are significant risk factors for the development of NHL in PLWH, with an estimated 11- to 17-fold increased risk in the late cART era compared to the general population [3,4]. HIV-associated NHLs typically present with advanced-stage disease, extranodal involvement, B symptoms, and aggressive clinical behavior [1]. Despite improvements in HIV care, NHL remains a leading cause of mortality among PLWH, underscoring the ongoing need for evidence-based therapeutic strategies [147].

DLBCL accounts for 30–50% of NHL cases in PLWH and is particularly prevalent in individuals with severe immunodeficiency (CD4^+^ T cell count < 200/µL) [4,164]. Prospective studies and pooled analysis have demonstrated that immunochemotherapeutic regimens incorporating cART and the anti-CD20 monoclonal antibody rituximab (R) are both safe and effective, yielding higher CR rates, longer progression-free survival (PFS), and improved overall survival (OS) in HIV-associated DLBCL [165,166,167,168,169]. Enhanced supportive care, particularly the prophylaxis of opportunistic infections, is critical in high-risk patients [1,150,152], especially in light of findings from a randomized phase 3 trial conducted in the early cART era, which identified safety concerns in patients with advanced HIV [170]. The most commonly used regimens include R-CHOP (cyclophosphamide, doxorubicin, vincristine, prednisone), infusional dose-adjusted (DA)-R-EPOCH (etoposide, prednisone, vincristine, doxorubicin, cyclophosphamide dose-adjusted to CD4 count), and infusional R-CDE (cyclophosphamide, doxorubicin, etoposide). In phase 2 trials, the 2-year OS rates were 75% for R-CHOP, 70% for DA-R-EPOCH, and 64% for R-CDE [165,166,167,168,169] (Table 2). A pooled analysis from two consecutive AIDS Malignancy Consortium studies indicated that infusional R-EPOCH may provide superior outcomes compared to bolus R-CHOP [171]. However, a randomized phase 3 trial in immunocompetent DLBCL patients showed that DA-R-EPOCH did not significantly improve outcomes compared to R-CHOP, suggesting that its benefit may be context-dependent [172].

In the randomized Phase 2 study AMC-075, the efficacy of dose-adjusted R-EPOCH alone was compared to its combination with the oral histone deacetylase inhibitor, oncolytic vorinostat, in patients with HIV-associated aggressive B-cell NHL (approximately 70% DLBCL. The addition of vorinostat conferred no significant benefit in terms of treatment outcomes or reduction of the HIV reservoir [173]. Notably, Myc protein expression was the only biomarker significantly associated with inferior outcomes: the 3-year event-free survival (EFS) of 44% in Myc-positive versus 83% in Myc-negative DLBCL, highlighting the urgent need for novel targeted therapies in this high-risk subgroup [173] (Table 2). A retrospective European study further suggested that Myc-driven HIV-associated DLBCL may benefit more from intensive chemotherapy regimens traditionally used for BL compared to standard R-CHOP, although these findings require validation in prospective clinical trials [174].

The prognostic significance of tumor histogenesis and molecular subtypes, such as double-hit and triple-hit B-cell lymphomas, remains controversial in HIV-associated DLBCL [1]. This uncertainty likely reflects the heterogeneity of study populations and methodological differences across investigations [143]. In immunocompetent patients, recent advances—particularly the use of the antibody-drug conjugate polantuzumab vedotin—have demonstrated a survival advantage over R-CHOP [175]. However, data on the efficacy and safety of such novel agents in HIV-associated DLBCL are currently lacking, underscoring the need for dedicated studies in this population.

PLWH with moderate immunodeficiency have a lifelong risk of developing BL estimated at 10–20%, regardless of cART status. This risk correlates more strongly with cumulative HIV viremia than with recent immunodeficiency [3,176]. Outcomes for HIV-associated BL have improved in the late cART era with the use of immunochemotherapy regimens that include cART and intensive or infusional multi-agent therapies; notably, R-CHOP is considered inadequate for the treatment of BL [141,177,178]. A phase 2 trial of modified CODOX-M/IVAC-R (cyclophosphamide, doxorubicin, vincristine, methotrexate, etoposide, ifosfamide, cytarabine, rituximab) demonstrated a 2-year overall survival of 69%, with a favorable toxicity profile [177,179]. In patients without central nervous system involvement, risk-adjusted DA-EPOCH-R has shown high efficacy regardless of HIV status, achieving a 4-year EFS rate of 85% [179,180] (Table 2). Furthermore, a large international retrospective study reported that CODOX-M/IVAC-R was associated with significantly longer PFS (hazard ratio (HR) 0.45, *p* = 0.005), improved OS (HR 0.44, *p* = 0.007), and lower treatment-related mortality (TRM) (7% vs. 13–18%) compared to other regimens, supporting its role as a preferred option for HIV-associated BL [181].

**Table 2 cancers-17-02088-t002:** Major front-line clinical trials in HIV-associated non-Hodgkin Lymphoma (HIV-NHL) and classical Hodgkin Lymphoma (HIV-cHL).

Treatment	Study Design	Patients N°	Histology %	CR Rate %	PFS %	OS %	I.Death %	Ref.
**HIV-NHL**								
R-CHOP-R vs. CHOP	Phase 3	150	DLBCL 81	58 vs. 47	11.3 vs. 9.5 mos	28 vs. 35	14 vs. 2	[170]
R-CHOP	Phase 2	61	DLBCL 72	77	69 (2 yr)	75 (2 yr)	2	[166]
R-CHOP	Phase 2	95	DLBCL 81	69	NA	56 (3 yr)	7	[169]
R-CDE	Phase 2	74	DLBCL 72	70	EFS (2 yr) 52	64 (2 yr)	7	[165]
R-EPOCH vs. EPOCH-R	Rand. Phase 2	106	DLBCL 80	73 vs. 55	66 vs. 63 (2 yr)	70 vs. 67 (2 yr)	10 vs. 7	[167]
SC-EPOCH-RR	Phase 2	33	DLBCL 100	91	84 (5 yr)	68 (5 yr)		[168]
Vorinostat-R-EPOCH	Rand. Phase 2	90	DLBCL 100	58 vs. 74	EFS (3 yr) 63 vs. 69	70 vs. 77 (3 yr)	NA	[173]
R-CODOX-M/IVAC	Phase 2	34	BL 100	NA	69 (1 yr)	69 (2 yr)	3	[177]
DA-EPOCH-R	Phase 2	28 HIV	BL 100	NA	85 (4 yr)	NA	NA	[179]
**HIV-cHL**								
ABVD	Retrospective	93	cHL	74	EFS (5-yr) 59	81 (5 yr)	<1	[182]
(a) EF: ABVD (b) EU: BEACOPP (c) Adv: BEACOPP or ABVD	Phase 2	108	cHL	(a) EF: 96 (b) EU: 100 (c) Adv: 86	90 (2 yr)	91 (2 yr)	EF: 4 Adv: 7	[183]
BV-AVD	Phase 2	41	cHL	76	86 (2 yr)	92 (2 yr)	<1	[184]

To date, there is no universally accepted standard of care for PBL and PEL in PLWH. Prognosis remains poor in population-based HIV studies, with 5-year OS rates below 40% for PBL and below 30% for PEL [1,142,185]. Nevertheless, favorable outcomes have been reported in selected patients treated with anthracycline-based multiagent chemotherapy in combination with cART [1,145,185]. In the AMC-075 trial, 3-year EFS rates for PBL and PEL were 60% and 71%, respectively [173]. A large international cohort study of patients with HIV-associated PEL treated with anthracycline-based chemotherapy, cART, and rituximab (when concurrent MCD was present) identified poor performance status and severe anemia as adverse prognostic factors. Notably, the median OS was 16.9 years in patients without these, compared to just 0.6 years in those with them—highlighting the urgent need for improved treatment strategies in HIV-associated PEL with poor prognosis [144]. Novel therapeutic approaches—including bortezomib, daratumumab, lenalidomide, nivolumab, and pomalidomide—are currently under investigation. Despite therapeutic advances, KSHV-associated large B-cell lymphomas remain particularly challenging, with persistently poor long-term outcomes [1].

Prospective and retrospective studies have shown that high-dose chemotherapy (HDC) followed by autologous stem cell transplantation (ASCT) is safe and effective in PLWH who have chemosensitive relapsed or refractory lymphoma, with TRM rates below 5% and 3-year overall survival ranging from 61% to 85% [186,187,188]. In contrast, data on allogeneic hematopoietic cell transplantation (allo-HCT) in PLWH are limited [189]. Nonetheless, for eligible PLWH, identifying a matched donor homozygous for the CCR5Δ32 mutation remains of particular interest, given reported cases of sustained HIV remission or virologic cure following allo-HCT with such donors [190,191]. Despite promising early data, PLWH continue to be excluded from most clinical trials investigating chimeric antigen receptor (CAR) T-cell therapy, representing an important unmet need in this population [192] (see below).

### 4.2. Current Treatment Strategies for HIV-Related Classic Hodgkin Lymphoma

PLWH who develop cHL typically present with moderate immunodeficiency, B symptoms, and advanced-stage disease involving the bone marrow, liver, and spleen. Primary bone marrow involvement occurs in approximately 3–14% of cases and is associated with particularly aggressive behavior [1,3]. As in the general population, stage-adapted therapy is recommended for HIV-associated cHL. The ABVD regimen (doxorubicin, bleomycin, vinblastine, dacarbazine) has demonstrated safety and efficacy in PLWH receiving cART, achieving a CR rate of 74% and a 5-year OS of 81% [183,193]. The BEACOPP regimen (bleomycin, etoposide, doxorubicin, cyclophosphamide, vincristine, procarbazine, and prednisolone) has also shown high efficacy, with a CR rate of 86%, but it is associated with considerable toxicity, including dose reductions or treatment delays in more than 50% of patients and a TRM of 6% [183] (Table 2).

Risk-adapted therapy guided by fluorodeoxyglucose positron emission tomography (FDG-PET) may serve as a standardized approach in HIV-associated cHL, mirroring strategies used in the general population [194]. However, caution is warranted, as non-malignant HIV-associated conditions can lead to false-positive PET-avid regions, particularly in patients with low CD4^+^ T cell counts.

The phase 2 AMC-085 trial evaluated the combination of the anti-CD30 antibody–drug conjugate brentuximab vedotin (BV) with AVD (doxorubicin, vincristine, dacarbazine) in newly diagnosed HIV-associated cHL. The study demonstrated promising outcomes, with 2-year PFS and 2-year OS rates of 86% and 92%, respectively, and manageable toxicity [176] (Table 2). Programmed death receptor (PD)-1 blockade has shown efficacy in cHL generally [195], but the historical exclusion of PLWH from oncology clinical trials has delayed the integration of immune checkpoint inhibitors (ICIs) into HIV-associated cHL treatment. Emerging data support the safety and efficacy of ICIs in PLWH with advanced malignancies [196,197,198]. Notably, the multicenter phase 3 SWOG 1826 trial, which included a small cohort of PLWH with well-controlled HIV infection, demonstrated that nivolumab, an anti-PD-1 monoclonal antibody, combined with AVD in newly diagnosed advanced cHL significantly improved PFS compared to BV-AVD in newly diagnosed advanced-stage cHL (HR 0.48, 95% CI 0.27–0.87, *p* < 0.001) and was associated with better tolerability [199]. Although these data need to be confirmed in a large series of HIV-cHL, they could accelerate the harmonization of treatment between HIV-infected and uninfected patients with cHL.

HDC, followed by ASCT, remains the standard of care for relapsed/refractory cHL, even in PLWH. For patients who experience relapse after HDC-ASCT, treatment with ICIs, such as nivolumab or pembrolizumab, is recommended, in line with standard practice in the general population [195]. However, the use of ICIs in PLWH may be complicated by the reactivation or exacerbation of latent infections, including Mycobacterium tuberculosis, hepatitis B virus, and cytomegalovirus [200,201]. Therefore, comprehensive microbiological screening, along with appropriate prophylaxis or treatment for opportunistic infections, is critical to mitigate the risk of infection-related adverse events in this population.

### 4.3. Current Treatment Strategies for HIV-Related Multicentric Castleman Disease

MCD is a relapsing and remitting lymphoproliferative disorder characterized by systemic inflammatory symptoms, edema or effusions, cytopenias, generalized lymphadenopathy, and splenomegaly. Active disease is associated with lytic KSHV replication and dysregulation of cytokines, with elevated levels of vIL-6, human IL-6, and IL-10 [92]. A recently described “fluid form” of MCD has been identified using multiparametric cytometry, where the presence of LANA-positive, lambda-restricted plasmablasts in effusions or peripheral blood samples may be indicative of MCD [202,203]. Nonetheless, lymph node biopsy remains essential for a definitive diagnosis of MCD. More than half of PLWH diagnosed with MCD also have concurrent Kaposi Sarcoma, and 20–35% may develop PEL or KSHV-associated large B cell lymphoma at diagnosis or during the disease course [144].

If left untreated, MCD is typically fatal. However, several prospective and retrospective multicenter studies have demonstrated that rituximab significantly improves survival, with reported 5-year OS rates between 80% and 95% [1,204]. In cases of severe disease, the combination of rituximab with cytotoxic agents such as etoposide or liposomal doxorubicin (particularly when Kaposi sarcoma is present) is recommended. In a U.S. cohort, PLWH with MCD treated with cART, rituximab, and liposomal doxorubicin had 5- and 10-year OS rates of 90% and 73%, respectively [129]. Antiviral therapy and IL-6 blockade have demonstrated limited clinical benefit in this setting [1,205]. Novel targeted therapies, including the anti-CD38 monoclonal antibody daratumumab and the JAK-STAT pathway inhibitor pacritinib, are currently under investigation [144]. Early recognition is critical, as misdiagnosis and delayed diagnosis of KSHV-associated diseases can significantly impair timely, disease-specific treatment and negatively impact patient outcomes [206].

## 5. Immunotherapeutic Perspectives for Lymphomas in PLWH

Despite the complex spectrum of immune dysfunctions associated with HIV infection, accumulating evidence indicates that a variety of immune-based therapies can be safely and effectively used to treat cancer in PLWH. These include monoclonal antibodies, immunomodulatory agents and cytokines, allo-HSCT, and cell-based therapies. Notably, cART itself serves as a critical form of immunotherapy for certain HIV-associated cancers such as Kaposi sarcoma and PCNSL. The immune reconstitution achieved through cART provides a foundational prerequisite for the success of any type of immunotherapy and remains a cornerstone of HIV-related cancer management. However, access to novel immunotherapies for lymphoma in PLWH has been limited, largely due to socioeconomic disparities and a historical lack of data on safety and efficacy in this population. Increasing evidence now supports the safety and clinical benefit of various immunotherapeutic approaches in PLWH, leading to growing advocacy for their inclusion in cancer clinical trials. These efforts are crucial for ensuring equitable access to innovative treatments and improving outcomes in this historically underrepresented group.

ICIs, particularly those targeting the PD-1/PD-L1 and CTLA-4 pathways, have demonstrated substantial clinical benefit across a range of malignancies in the general population, particularly in tumors characterized by high mutation burden and pre-existing anti-tumor immunity. These agents are particularly promising for PLWH, as they function by reversing immune exhaustion rather than augmenting immune suppression. PLWH typically exhibit elevated levels of exhausted CD8^+^ and CD4^+^ T cells that express multiple inhibitory checkpoint molecules, suggesting a potential heightened responsiveness to ICI [207]. Beyond their anticancer effects, ICIs may influence HIV persistence. Several studies have shown that PD-1 blockade can enhance HIV transcription and replication from latently infected cells, potentially facilitating immune recognition and clearance of HIV-infected cells that persist on cART. In a study of 32 PLWH with cancer receiving cART, anti-PD-1 treatment was associated with in vivo reversal of HIV latency, supporting a rationale for combining ICIs with latency-reversing agents or other HIV-curative strategies [208]. However, results have been variable: the AIDS Malignancy Consortium 095 Study found that anti-PD-1 monotherapy had no significant impact on HIV latency or reservoir size. In contrast, dual checkpoint blockade with anti-PD-1 and anti-CTLA-4 modestly increased levels of cell-associated unspliced HIV RNA, suggesting potential synergy in targeting the replication-competent HIV reservoir [209].

However, the use of ICIs in patients with HIV-related lymphomas has historically been limited due to safety concerns, particularly the potential for disrupting immune homeostasis and triggering immune-related adverse events (irAEs) or chronic inflammatory syndromes. However, emerging evidence has provided reassurance regarding the safety profile of ICIs in PLWH. Recent studies report that severe irAEs (grade ≥ 3) occur in approximately 0–20% of treated individuals, with most patients maintaining stable HIV viral loads and CD4^+^ T-cell counts throughout treatment [210,211,212,213,214,215]. Notably, the prospective, real-world ANRS CO24 OncoVIHAC cohort study, which evaluated ICI therapy in PLWH with cancer, reported a 1-year incidence of a first grade ≥ 3 irAE episode of 15.0%, with a cumulative incidence of all severe irAE episodes of 26.9 per 100 person-years. The study also identified several risk factors for the development of severe irAEs, including low CD4^+^ T-cell count, longer duration since HIV diagnosis, history of cancer surgery, and positive cytomegalovirus serology at the initiation of ICI therapy [215].

The assessment of the therapeutic efficacy of ICIs in PLWH and cancer remains limited by the small number of patients enrolled in clinical trials. While ICIs have demonstrated modest benefits in immunocompetent patients with aggressive NHLs, the expression of viral antigens encoded by EBV and/or KSHV in certain HIV-related lymphomas may enhance their immunogenicity, potentially increasing responsiveness to ICI therapy [213,214]. This aligns with the broader observation that ICI efficacy is generally higher in patients with more immunogenic tumors that elicit pre-existing anti-tumor immune responses. As noted above, available clinical data suggest that anti-PD-1 therapy can be effective in HIV cHL [215], as well as in NHLs driven by oncogenic viruses in PLWH [213,214]. Despite these encouraging findings, the proportion of PLWH and cancer who respond to ICI therapy remains limited, mirroring outcomes in the general population. This underscores the need for novel therapeutic strategies to enhance ICI efficacy and broaden their applicability in this setting. One promising approach involves the use of rational drug combinations that synergize with ICIs. A retrospective study of a small cohort of PLWH with lymphomas evaluated pembrolizumab alone or in combination with pomalidomide, an immunomodulatory agent with both direct anti-lymphoma activity and the ability to enhance T and NK cell activation. Pomalidomide also reverses EBV- and KSHV-mediated immune evasion by restoring the expression of MHC I, ICAM-1, and CD86 on tumor cells [216,217]. A response rate of 50% was observed, with notable activity against PEL, suggesting that this combination merits further investigation [217,218]. Advances in understanding the immune landscape of HIV-related lymphomas are also guiding the development of mechanism-based ICI combinations. For example, the upregulation of the TIGIT ligand CD155 in neoplastic cells of HIV-associated cHL, along with increased expression of nectin-3, provides a compelling rationale for targeting the TIGIT/CD155/Nectin-3 axis. Clinical trials are currently underway to evaluate the therapeutic potential of disrupting this pathway in cHL and other malignancies [219].

Chimeric antigen receptor (CAR) T cell therapy has represented a major advancement in the treatment of relapsed or refractory DLBCL in the general population [220]. While five generations of CAR T cell products have been developed, all current FDA-approved products for hematologic malignancies are second-generation CAR T cells [221]. Given the persistently poor outcomes in PLWH and relapsed/refractory DLBCL despite therapeutic advances [164], CAR T cell therapy emerges as an especially attractive option for this population. However, PLWH were excluded from pivotal licensing trials of CAR T cell therapies, thereby delaying the application of this potentially effective approach to HIV-infected patients with lymphoma [222,223,224]. A recent review detailed the clinical characteristics and outcomes of the limited number of PLWH with DLBCL treated with CAR T cells [224]. In a small cohort, treatment with axicabtagene ciloleucel resulted in clinical responses in four of six patients (three complete remissions and one partial remission). Adverse events included cytokine release syndrome (CRS) of grade ≤ 2 and immune effector cell-associated neurotoxicity syndrome (ICANS) of grade 3–4 in four patients. Although preliminary, these findings suggest that CAR T cell therapy in PLWH may have an acceptable safety profile and therapeutic efficacy, with no compelling clinical contraindications identified to date. One of the key barriers to the broader application of CAR T cell therapy in PLWH is the challenge of achieving sufficient numbers of functional, particularly concerning CD4^+^ T cells. This lymphocyte subset is not only critical for the manufacturing of a clinical-grade product but also contributes to the therapeutic efficacy and efficacy of CAR T cells [225,226,227,228]. This issue may be particularly relevant in patients with advanced HIV disease or low CD4 counts at diagnosis. Nonetheless, several case reports have demonstrated that CAR T cell products can be successfully manufactured from patients with controlled HIV infection, even when CD4^+^ T cell counts are relatively low [226]. Additional concerns include the presence of low-level viremia or transient increases in viral load during cART, which is relatively common and generally does not reflect the loss of virologic control. These should not, therefore, be used as exclusion criteria for CAR T cell treatment in PLWH. Another consideration is the use of lentiviral vectors in CAR T cell manufacturing. In the context of HIV infection, this raises potential concerns regarding enhanced CAR transgene expression during HIV replication, as well as the risk of false-positive HIV test results due to vector components [229]. Consequently, gamma-retroviral vectors have thus far been preferred for manufacturing CAR T cell products in PLWH with lymphoma. In summary, although data remain limited, current evidence supports the feasibility, safety, and potential efficacy of CAR T cell therapy in PLWH with relapsed/refractory lymphoma. Future clinical applications in this setting may benefit from advances in next-generation CAR T cell technologies, including strategies to enhance T cell potency, overcome tumor heterogeneity and antigen escape, and improve safety and specificity [230]. Looking ahead, the rational combination of these innovations may allow CAR T cells to not only treat HIV-related malignancies but also potentially contribute to HIV cure strategies. This could be achieved through the development of multi-specific CAR T cells targeting tumor- and virus-associated antigens and engineered to be resistant to HIV infection via gene editing techniques [231].

Hematopoietic stem cell transplantation (HSCT) has demonstrated encouraging outcomes in the clinical management of lymphomas. In the context of HIV infection, particular attention has been given to allo-HSCT using donor stem cells carrying the CCR5Δ32 mutation, which confers resistance to HIV by preventing viral entry into host cells. This approach has successfully led to long-term HIV remission or apparent cure in five individuals with HIV-associated hematologic malignancies [232]. Notably, a recent case report showed that sustained HIV remission can also be achieved following allogeneic HSCT from a donor without the CCR5Δ32 mutation, indicating that this genetic deletion is not an absolute requirement for achieving long-term viral control or potential cure [233]. These findings are highly significant, as they suggest a broader applicability of allo-HSCT to a greater number of HIV-infected patients with cancer, including lymphoma, potentially improving their clinical outcomes.

Adoptive cell therapy with EBV-specific T cells is also being explored as a potential strategy for treating EBV-associated malignancies in PLWH. This approach has already demonstrated clinical benefit in managing post-transplant lymphoproliferative disease (PTLD) and controlling EBV reactivation in immunocompromised hosts [234]. In HIV-negative individuals with relapsed or refractory NHL or cHL, treatment with autologous cytotoxic T lymphocytes specific for EBV antigens LMP1/LMP2 or LMP2 alone yielded encouraging results, with most patients achieving 2-year EFS with minimal toxicity [235]. Similarly, infusion of autologous EBNA-1-specific T cells has been shown to restore EBV-specific immunity and control EBV-infected B cell expansion in both adult and pediatric patients with EBV viremia or PTLD following HSCT [236]. More recently, EBV-specific T cells engineered to resist the immunosuppressive activity of Transforming Growth Factor-β have been shown to safely expand and persist in patients with cHL, even in the absence of lymphodepleting chemotherapy, and have induced CR even in patients with resistant disease [237]. Despite these promising results, the application of adoptive EBV-specific T cell therapy in PLWH remains investigational. Challenges include T cell exhaustion and low precursor frequencies, which may hinder the generation of effective autologous EBV-specific T cells in HIV-positive individuals. To overcome these limitations, the use of allogeneic or third-party EBV-specific T cells is being considered [238].

## 6. Conclusions

Lymphomas in PLWH represent a highly heterogeneous group of malignancies that pose substantial challenges in cancer research, diagnosis, and treatment. These challenges stem from their complex pathogenesis, immunologically composite TME, and frequently aggressive clinical course. Despite notable progress in recent years, advancements in the diagnostic evaluation and therapeutic management of lymphomas in PLWH remain constrained by fragmented research efforts and the frequent exclusion of this population from pivotal clinical trials. Collaborative, multidisciplinary strategies are urgently needed to address these gaps, reduce the incidence of HIV-associated lymphoproliferative disorders, and ultimately improve clinical outcomes and survival for these patients.

## Figures and Tables

**Figure 1 cancers-17-02088-f001:**
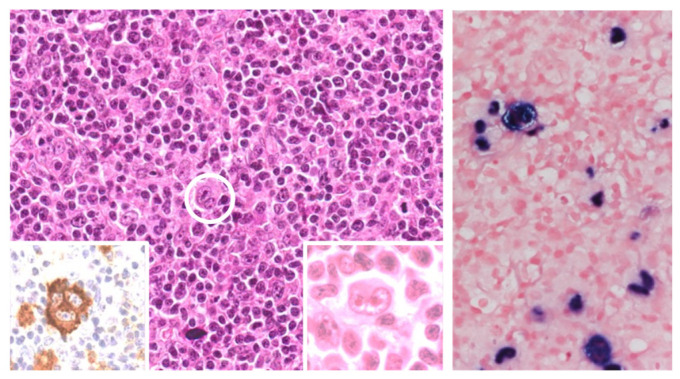
EBV-positive classic Hodgkin lymphoma (cHL) in a patient living with HIV. (**Left**) Hodgkin–Reed–Sternberg (HRS) cells (circled area) are seen in an inflammatory environment. A diagnostic HRS cell is encircled at the center, expresses CD30 antigen (in the inset to the left), and is rosetted by small lymphocytes (inset to the right). (**Right**) EBER in situ hybridization shows EBV positivity in sparse tumor cells. (**Left**) Hematoxliyn and Eosin stain, original magnification 40×, Inset to the left: immunohistochemistry, inset to the right: Hematoxliyn and Eosin; Inset: original magnification 100×; (**Right**) in situ hybridization, original magnification 40×.

**Figure 2 cancers-17-02088-f002:**
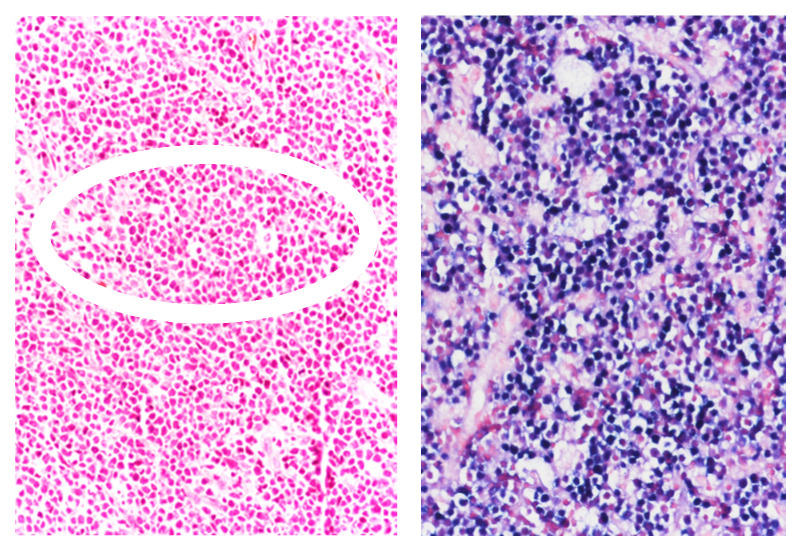
EBV-positive Burkitt lymphoma in a patient living with HIV. (**Left**) the tumor population is monotonous and displays small-intermediate size. Starry sky pattern is seen at the center of the microphotograph (indicated by an ellipse). (**Right**) EBER in situ hybridization shows EBV positivity in most tumor cells. (**Left**) Hematoxliyn and Eosin stain, original magnification 20×, (**Right**) in situ hybridization, original magnification 20×.

**Figure 3 cancers-17-02088-f003:**
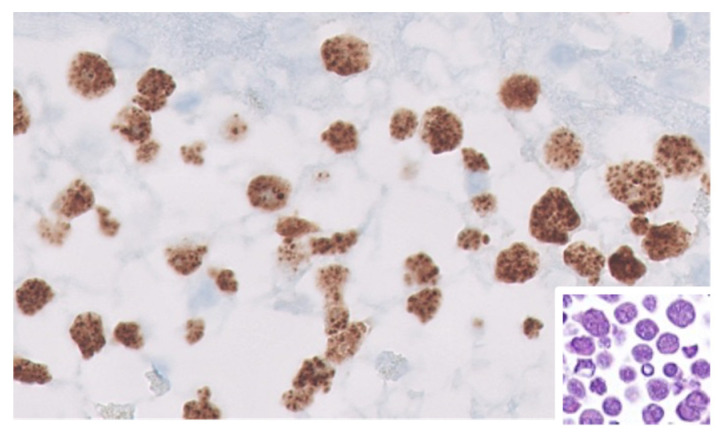
Kaposi sarcoma herpesvirus (KSHV)/human herpes virus 8 (HHV8)-positive primary effusion lymphoma (PEL) in a patient living with HIV. Sparse tumor cells displaying various sizes show nuclear positivity for latent nuclear antigen (LNA) of KSHV/HHV8. The staining is characteristically “speckled”. The inset shows that tumor cells are anaplastic and usually display large size. Immunohistochemistry for LNA, original magnification 40×. Inset: Hematoxliyn and Eosin stain from a cell block of a pleural effusion, original magnification 40×.

**Table 1 cancers-17-02088-t001:** Lymphoid proliferations associated with oncogenic virus infections in people with immunodeficiency [1,81,91,92].

Type of Lymphoproliferation	Viral Status (EBV and KSHV)
B-cell hyperplasias	EBV+; KSHV−
Polymorphic proliferations	EBV+ or −
Indolent B-cell lymphomas	EBV+/−; KSHV−
B-cell lymphomas *	
Hodgkin lymphoma	EBV+; KSHV−
Burkitt lymphoma	EBV−/+; KSHV−
DLBCL	EBV+; KSHV−
PEL and its solid variants	EBV+; KSHV+
PBL of the oral cavity type	EBV+/−; KSHV−
MCD-associated LBCL	EBV−; KSHV+
PBL associated with MCD	EBV−; KSHV+

Immunodeficiency setting includes post-transplant (solid organ), iatrogenic (methotrexate), immune senescence, and people living with HIV/AIDS. * Must meet criteria for corresponding lymphoma in immunocompetent hosts.

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
