# Peer review of "Lymphoproliferations in People Living with HIV: Oncogenic Pathways, Diagnostic Challenges, and New Therapeutic Opportunities"

_cancers, 2025, doi:10.3390/cancers17132088_

Round 1

Reviewer 1 Report

Comments and Suggestions for Authors

Review: General comments

This is a thorough review of the current state of lymphoproliferative disorders in people living with HIV(PLWH). As the longevity gap in PLWH decreases, and people live longer, lymphomas become important causes of morbidity and mortality.  This review  includes clinical features, pathogenesis, including molecular mechanisms, histology and treatment challenges. There is an exploration of immunotherapy to enhance effective treatment in the setting of HIV. The bringing together of molecular pathogenesis and established and emerging treatment options is useful in identifying areas of further research to optimise treatments for virally suppressed PLWH in the era of cART so that treatment options and outcomes match those of the general population..

Overall, the topic is well-covered, with a thorough review of the literature.  

Please find detailed comments below.

Specific comments:

I would recommend the authors address the following aspects to enhance completeness and readability:

  1. Page 2, line 40: With the widespread use of combination antiretroviral therapy……within many developed countries. Provide a reference/s.

  1. Page 2, line 46-47: The 5th edition of the World Health Organization classification of lympho-hemopoietic neoplasms (WHO-HAEM5) .Replace with hematolymphoid tumors. (5th edition WHO, English)

  1. Page 2, lines71-74: HIV-induced depletion of helper CD4+ T cells is associated with increased expression of the inhibitory checkpoint molecules PD-1, LAG3, TIM-3 72 and TIGIT in CD8+ T cells, leading to their functional exhaustion and impaired control of emerging tumor cells. Reference 7 states the following: “Our data provide evidence that CD4+ T cells expressing PD-1, TIGIT and LAG-3 alone or in combination are enriched for persistent HIV during ART and suggest that immune checkpoint blockers directed against these receptors may represent valuable tools to target latently infected cells in virally suppressed individuals.” Please clarify this statement.

  1. Page 4, line 148: latency-associated nuclear antigen – please write out in full the first time an abbreviation is used here and everywhere else within the article.

  1. Page 4, Line 167-170: While cART improves overall immune function ……..development of lymphoproliferative disorders. Please provide a reference.

6. Page 5, line 195: AICDA: Activation-induced cytidine deaminase; write out in full the first time .

  1. Page 5, line : 211: HIV-1 matrix protein p17 – write out in full for the first time.

8. Page 6, line 278 : Table 1: I would also specifically mention plasmablastic lymphoma associated with multicentric Castleman's disease - EBV negative/KSHV positive.

9. Page 7 , line 293-294: Outgrowth may read better as overgrowth

  1. Page 7, line 298: Authors state that cART increases risk of HL to 20x that of the general population. Ref 86 states: However, the incidence of HL declines after the first year of cART, suggesting immune reconstitution may be linked to HL development. This elevated risk does not persist after the first year.

   Whist the decline with ART is not quite that seen with NHL I would temper or clarify this statement as it creates an erroneous impression of persistently elevated risk despite ART.

Additional facts: the risk of HL among HIV patients with controlled viral load and CD4 recovery on cART remains 9-fold higher than the general population (Hleyhel et al. 2014), suggesting that recovery of CD4 cells is initially insufficient to control elevated risk for HL after HIV infection; however, the incidence of HL declines after the first year of cART, suggesting immune reconstitution may be linked to HL development.  

  1. Page 9, line 372-374 : I would add the following to the characteristic clinical features of BL:

11.1There is a strong correlation with cumulative viraemia.

11.2 Intraoral lesions are common and there is a high risk of CNS involvement.

  1. Page 9, Line 376, (Figure 2) – this is not anchored to anything. Perhaps add the following: The histological features and EBV staining are illustrated in Figure 2 below.

  1. Page 9, Line 377-379: Authors state: The HIV Tat protein was detected within the tumor cells of BL patients….. This is in the past tense then followed by sentences in the present tense. Maintain the correct tense throughout the article.

  1. Page 9, line 395-400. Whist acknowledging that plasmablastic lymphoma is uncommon in developed setting, they are certainly more common in high HIV prevalence countries and there are some established facts about clinical presentation which can be included:

All PBL are CD20 negative. 75% are EBV positive. There is a definite male predominance, most cases are extra-nodal, especially with head and neck lesions. followed by skin, GIT, and bone. Median survival 6-12 months. The molecular pathogenesis is characterized by a dysregulation of MYC, a key oncogene, and activation of pathways like JAK-STAT, RAS-RAF, and NOTCH etc.

  1. Page 10, below 431: The authors make no mention of T cell lymphomas in PLWH. They are uncommon but a very brief mention of incidence, types, and mention of HTLV-1 virus for adult T-cell leukemia/lymphoma (ATLL) and also association with EBV is recommended.

  1. Page 12, Lines 518-524 – this deals with the clinical features pf PBL and would fit better in that section. It is not a discussion of treatment.
Comments on the Quality of English Language

This paper would benefit from a general edit to streamline and pare down sentences and iron out grammatical gremlins.  

Author Response

Response point by point to Reviewer 1

We thank the reviewer for the insightful comments and suggestions that have been helpful to improve clarity and readability of the manuscript.

  1. Page 2, line 40: With the widespread use of combination antiretroviral therapy……within many developed countries. Provide a reference/s.

According to the Reviewer’s suggestion, the following references have been included:

-Reference n°1 (numbered previously as n°127):

Ramaswami R, Chia G, Dalla Pria A, et al. Evolution of HIV-Associated Lymphoma Over 3 Decades. J Acquir Immune Defic Syndr. 2016;72(2):177-183. Doi:10.1097/QAI0000000000000946

-Reference n°2 (numbered previously as n°153):

Hernández-Ramírez RU , Shiels MS , Dubrow R , Engels EA. Cancer risk in HIV-infected people in the USA from 1996 to 2012: a population-based, registry-linkage study. Lancet HIV 2017 Nov;4(11):e495-e504.  doi: 10.1016/S2352-3018(17)30125-X. 

 - Reference n°3 (numbered previously as n°154):

Kimani SM , Painschab MS , Horner MJ , Muchengeti M , Fedoriw Y , Shiels MS , Gopal S. Epidemiology of haematological malignancies in people living with HIV. Lancet HIV 2020 Sep;7(9):e641-e651.  doi: 10.1016/S2352-3018(20)30118-1.

  Reference n° 4: (numbered previously as n°1):

Carbone A, Vaccher E, Gloghini A. Hematologic cancers in individuals infected by HIV. Blood. 2022;139(7):995-1012. Doi:10.1182/blood.2020005469

  1. Page 2, line 46-47: The 5th edition of the World Health Organization classification of lympho-hemopoietic neoplasms (WHO-HAEM5). Replace with hematolymphoid tumors. (5thedition WHO, English)

The reference has been corrected as suggested by the Reviewer.

  1. Page 2, lines71-74: HIV-induced depletion of helper CD4+ T cells is associated with increased expression of the inhibitory checkpoint molecules PD-1, LAG3, TIM-3 72 and TIGIT in CD8+ T cells, leading to their functional exhaustion and impaired control of emerging tumor cells. Reference 7 states the following: “Our data provide evidence that CD4+ T cells expressing PD-1, TIGIT and LAG-3 alone or in combination are enriched for persistent HIV during ART and suggest that immune checkpoint blockers directed against these receptors may represent valuable tools to target latently infected cells in virally suppressed individuals.” Please clarify this statement.

We thank the reviewer for the opportunity to clarify this point. The sentence referred to CD4 T cells as reservoir of HIV. We have changed this sentence by stating: “Specifically, the sustained expression of inhibitory checkpoint receptors, such as PD-1, TIM-3 or TIGIT on T cells limits their cytotoxic potential against emerging tumor cells [10]”. A new, pertinent reference was included: Benito JM, Restrepo C, García-Foncillas J, Rallón N. Immune checkpoint inhibitors as potential therapy for reverting T-cell exhaustion and reverting HIV latency in people living with HIV. Front Immunol. 2023 Dec 7;14:1270881. doi: 10.3389/fimmu.2023.1270881

  1. Page 4, line 148: latency-associated nuclear antigen – please write out in full the first time an abbreviation is used here and everywhere else within the article.

As suggested by the reviewer, LANA was written in full the first time the abbreviation is used and all other abbreviations have been thoroughly checked.

  1. Page 4, Line 167-170: While cART improves overall immune function ……..development of lymphoproliferative disorders. Please provide a reference.

As suggested by the Reviewer, two new references have been included:

  • Deeks SG. HIV infection, inflammation, immunosenescence, and aging. Annu Rev Med 2011:62:141-55.  doi: 10.1146/annurev-med-042909-093756.
  • Straista M, Caccuri F , Arnaut N , Caruso A , Slevin  Pathological Mechanisms Involved in HIV-Associated Lymphomagenesis: Novel Targeted Therapeutic Approaches. Cells 2025 May 13;14(10):705.  doi: 10.3390/cells14100705

  1. Page 5, line 195: AICDA: Activation-induced cytidine deaminase; write out in full the first time .

    AICDA was written in full on page 4, line 143 of the original manuscript.

  1. Page 5, line : 211: HIV-1 matrix protein p17 – write out in full for the first time.

As suggested by the Reviewer, HIV-1 matrix protein p17 was written out in full for the first time

  1. Page 6, line 278 : Table 1: I would also specifically mention plasmablastic lymphoma associated with multicentric Castleman's disease - EBV negative/KSHV positive.

As suggested by the Reviewer, Table 1 was completed by adding plasmablastic lymphoma associated with multicentric Castleman's disease

  1. Page 7 , line 293-294: Outgrowth may read better as overgrowth.

As suggested by the Reviewer, we have replaced the word “outgrowth” with “overgrowth”.

  1. Page 7, line 298: Authors state that cART increases risk of HL to 20x that of the general population. Ref 86 states: However, the incidence of HL declines after the first year of cART, suggesting immune reconstitution may be linked to HL development. This elevated risk does not persist after the first year.

   Whist the decline with ART is not quite that seen with NHL I would temper or clarify this statement as it creates an erroneous impression of persistently elevated risk despite ART.

Additional facts: the risk of HL among HIV patients with controlled viral load and CD4 recovery on cART remains 9-fold higher than the general population (Hleyhel et al. 2014), suggesting that recovery of CD4 cells is initially insufficient to control elevated risk for HL after HIV infection; however, the incidence of HL declines after the first year of cART, suggesting immune reconstitution may be linked to HL development.

We thank the Reviewer for the insightful comment. We have revised the text to improve clarity: “In high-income countries, PLWH have a 5- to 26-fold increased risk of developing HL compared to the general population [4]. The highest risk period for HL diagnosis often coincides with the months following initiation of cART, during which the inflammatory IRIS may occur [104,105]. Unlike NHL, the incidence of HL in PLWH has remained relatively stable or has declined only modestly in the cART era [3,4]. Notably, PLWH on effective cART - with suppressed viral load and restored CD4+ T cell counts - still exhibit a ninefold higher risk of HL than HIV-negative individuals [106], further implicating IRIS and immune dysregulation as potential contributors to HIV-associated HL pathogenesis [105,107]”.

-New References:

-Kowalkowski MA , Mims MP, Amiran ES,  Lulla P, Chiao EY. Effect of immune reconstitution on the incidence of HIV-related Hodgkin lymphoma. PLoS One 2013 Oct 2;8(10):e77409.  doi: 10.1371/journal.pone.0077409. eCollection 2013.

-Hleyhel M , Bouvier AM,  Belot A, Tattevin P, Pacanowski J, Genet P,  De Castro N, Berger JL, DupontC, Lavolé A, Pradier C, Salmon D, Simon A,  Martinez V, Spano JP, Costagliola D, Grabar S; Cancer Risk Group of the French Hospital Database on HIV (FHDH-ANRS CO4). Risk of non-AIDS-defining cancers among HIV-1-infected individuals in France between 1997 and 2009: results from a French cohort. AIDS 2014 Sep 10;28(14):2109-18.  doi: 10.1097/QAD.0000000000000382.

  1. Page 9, line 372-374 : I would add the following to the characteristic clinical features of BL:

11.1There is a strong correlation with cumulative viraemia.

11.2 Intraoral lesions are common and there is a high risk of CNS involvement.

In agreement with reviewer’s suggestions, we have revised the initial paragraph of HIV-associated BL as follows:

BL accounts for approximately 25-40% of lymphomas occurring in PLWH [1-3]. Despite the introduction of cART, the incidence of HIV-associated BL has not declined [1-4]. PLWH with moderate immunodeficiency carry a lifelong risk of 10-20% for developing BL, a risk that is more strongly associated with cumulative HIV viremia than with recent immune status [3,4]. Clinically, HIV-associated BL presents with distinct features compared to BL in HIV-negative individuals [123]. Patients often present with B symptoms, advanced-stage disease and frequent extranodal involvement - including bone marrow, oral cavity, gastrointestinal tract, and central nervous system [1]”.

  1. Page 9, Line 376, (Figure 2) – this is not anchored to anything. Perhaps add the following: The histological features and EBV staining are illustrated in Figure 2 below.

The text has been revised accordingly

  1. Page 9, Line 377-379: Authors state: The HIV Tat protein was detected within the tumor cells of BL patients….. This is in the past tense then followed by sentences in the present tense. Maintain the correct tense throughout the article.

The English language has been thoroughly revised throughout all the manuscript.

  1. Page 9, line 395-400. Whist acknowledging that plasmablastic lymphoma is uncommon in developed setting, they are certainly more common in high HIV prevalence countries and there are some established facts about clinical presentation which can be included:

All PBL are CD20 negative. 75% are EBV positive. There is a definite male predominance, most cases are extra-nodal, especially with head and neck lesions. followed by skin, GIT, and bone. Median survival 6-12 months. The molecular pathogenesis is characterized by a dysregulation of MYC, a key oncogene, and activation of pathways like JAK-STAT, RAS-RAF, and NOTCH etc.

We thank the reviewer for the pertinent suggestion. The section: “3.7. Other Rare HIV-Associated Lymphomas” has been revised accordingly with the inclusion of new references.

  1. Page 10, below 431: The authors make no mention of T cell lymphomas in PLWH. They are uncommon but a very brief mention of incidence, types, and mention of HTLV-1 virus for adult T-cell leukemia/lymphoma (ATLL) and also association with EBV is recommended.

We thank the reviewer for the pertinent suggestion. The section: “3.7. Other Rare HIV-Associated Lymphomas” of the revised manuscript now report a short description of the rare HIV-related T cell lymphomas.

  1. Page 12, Lines 518-524 – this deals with the clinical features pf PBL and would fit better in that section. It is not a discussion of treatment.

In agreement with Reviewer’s suggestion, the paragraph has been revised (see above, point 14)

Reviewer 2 Report

Comments and Suggestions for Authors

This is a comprehensive and scholarly review article that effectively synthesizes the current understanding of lymphoproliferative disorders in people living with HIV (PLWH). The manuscript integrates molecular pathogenesis, clinicopathological subtypes, tumor microenvironment, diagnostic features, and therapeutic approaches, including the evolving role of immunotherapies. The structure is logical and well-organized, and the citations are extensive and up-to-date.

While this is a timely and broadly informative review on HIV-associated lymphoproliferative disorders, the manuscript suffers from multiple significant issues that must be addressed prior to consideration for publication.

Major Weaknesses

  1. Lack of Source Attribution for Figures 1-3. If these are unpublished data, the authors need to disclose and provide ethical approval, patient consent, or reuse licensing.
  2. Structural and Organizational Issues. Certain sections (e.g., Pathogenesis) are confusing. For example, the subtitles 3.1 and 3.2 may be removed to make this section clearer; or re-number 3.3-3.7 to 3.2.1-3.2.5 to put them under section 3.2 since they are different “Lymphomas arising in PLWH”. Also, the authors should consider listing these lymphomas in two categories “HL and non-HL”.
  3. Redundancy: Tat, p17, and AICDA mechanisms are repeated across multiple sections (e.g., Tat’s role in MYC translocation is discussed in multiple paragraphs). EBV latency patterns (I–III) are explained at least twice in detail; consolidate into one clear explanation and refer back as needed.
  4. Section 2.1 and 3.2. The tumor microenvironment should be the main “indirect mechanism” and therefore this part of 3.2 should be integrated into 2.1.
  5. Section 3.3. “The probability of developing Hodgkin lymphoma (HL) in PLWH is 5-15 times higher than general population [85], and the use of cART increases this risk to about 20-30 times (than) that of the general population [86] (23)”. “The risk of HL remains nine times higher than in the general population [87].” These statements are very confusing. Please verify all the numbers carefully with appropriate citations. Does cART really increases this risk to 20-30 times? What does “The risk of HL remains nine times higher than in the general population” mean?
  6. Tat/p17 mechanisms and EBV synergy are scattered across multiple paragraphs. Important themes like the tumor microenvironment and immune escape are discussed in pieces across different subsections without cohesive integration.
  7. Line 167: “cART may induce metabolic changes, chronic inflammation, and immune dysregulation...” → Needs clarification: does cART cause these effects or fail to resolve them? References are needed for this claim.
  8. Lines 261-269. “They (Polymorphic lymphoproliferative disorders) are characterized morphologically by a heterogeneous, or polymorphic, cell population representing all stages of B cell differentiation”; “These polymorphic lymphoproliferative disorders usually contain monoclonal B cells and oligoclonal T cell populations”. Please clarify these statements, with appropriate citations.
  9. Lines 266-267: “A variable number of Reed-Sternberg-like cells are EBV+ and express the EBV oncogenic protein the latent membrane protein-1 (LMP1) cells”.
  10. Table 1. An additional column is needed for related references
  11. Include a table summarizing treatment options by lymphoma subtype, including regimens, outcomes, and key trials.

Minor issues

  • Line 59: “HIV my also directly contribute...” → “HIV may also directly contribute...”.
  • Line 126: “EBNA6 expressed by EBV-infected B cells may may lead...” → redundant “may”: “may lead”.
  • Lines 222-229. PI3K-Akt…. PI3K-Protein Kinase B.
  • Line 273: “based on identical morphologic, immunophenotypic and genotypic criteria as for lymphomas...” → Suggest: “based on the same morphologic, immunophenotypic, and genotypic criteria used in lymphomas...”.
  • Line 299: “use of cART increases this risk to about 20–30 times...” → Reference [86] does not clearly support this dramatic increase. Needs verification or rephrasing.
  • Section 3.5. Diffuse large B cell lymphoma (DLBCL). Since “DLBCL is the most common form of HIV-associated lymphoma”, it should be listed first.
  • Subtitles 5 Immunotherapeutic perspectives for lymphomas in PLWH; 5. Conclusions

In summary, the topic is important, and the literature synthesis is potentially valuable, but significant editorial, ethical, and scientific revisions are required. Major revision is required to bring the manuscript to the level expected for publication in Cancers.

Author Response

Response point by point to Reviewer 2

We thank the reviewer for the insightful comments and suggestions that have been helpful to improve clarity and readability of the manuscript.

Major Weaknesses 

  1. Lack of Source Attribution for Figures 1-3. If these are unpublished data, the authors need to disclose and provide ethical approval, patient consent, or reuse licensing.

Figures 1-3 are unpublished data, deriving from files of one of the Author (AC). The cases were seen at the Centro di Riferimento Oncologico (CRO) of Aviano and were submitted for publication with the ethical approval of the Institute and consent of the patients.

  1. Structural and Organizational Issues. Certain sections (e.g., Pathogenesis) are confusing. For example, the subtitles 3.1 and 3.2 may be removed to make this section clearer; or re-number 3.3-3.7 to 3.2.1-3.2.5 to put them under section 3.2 since they are different “Lymphomas arising in PLWH”. Also, the authors should consider listing these lymphomas in two categories “HL and non-HL”.

In agreement with the Reviewer, we have removed the subtitles in Section 3. A sentence has been included stating that ”.. and may be broadly identified as Hodgkin lymphomas (HL) and Non-Hodgkin lymphomas (NHL).”

  1. Redundancy: Tat, p17, and AICDA mechanisms are repeated across multiple sections (e.g., Tat’s role in MYC translocation is discussed in multiple paragraphs). EBV latency patterns (I–III) are explained at least twice in detail; consolidate into one clear explanation and refer back as needed.

In agreement with the reviewer, the entire text has been thoroughly revised to avoid repetitions and redundancies.

  1. Section 2.1 and 3.2. The tumor microenvironment should be the main “indirect mechanism” and therefore this part of 3.2 should be integrated into 2.1.

In agreement with the Reviewer, part of 3.2 has been integrated into 2.1

  1. Section 3.3. “The probability of developing Hodgkin lymphoma (HL) in PLWH is 5-15 times higher than general population [85], and the use of cART increases this risk to about 20-30 times (than) that of the general population [86] (23)”. “The risk of HL remains nine times higher than in the general population [87].” These statements are very confusing. Please verify all the numbers carefully with appropriate citations. Does cART really increases this risk to 20-30 times? What does “The risk of HL remains nine times higher than in the general population” mean?

We thank the Reviewer for the pertinent comment. We have revised this part in the new version of the manusctipt. Please, refer to the response to comment n° 10 of Reviewer 1.

  1. Tat/p17 mechanisms and EBV synergy are scattered across multiple paragraphs. Important themes like the tumor microenvironment and immune escape are discussed in pieces across different subsections without cohesive integration.

The text has been revised according to the Reviewer’s suggestion.

7. Line 167: “cART may induce metabolic changes, chronic inflammation, and immune dysregulation...” → Needs clarification: does cART cause these effects or fail to resolve them? References are needed for this claim.

In agreement with Reviewer comment, we have corrected that sentence: “While cART improves overall immune function and reduces HIV-related mortality, long-term use of cART in advanced HIV infection reduces but does not normalize the metabolic changes, chronic inflammation, and immune system dysregulation that  could indirectly contribute to the development of lymphoproliferative disorders (Deeks SG Annu Rev Med 2011; Straista M Cells 2025).”

  1. Lines 261-269. “They (Polymorphic lymphoproliferative disorders) are characterized morphologically by a heterogeneous, or polymorphic, cell population representing all stages of B cell differentiation”; “These polymorphic lymphoproliferative disorders usually contain monoclonal B cells and oligoclonal T cell populations”. Please clarify these statements, with appropriate citations.

In agreement with Reviewer comment, we have included additional references supporting the statements.

  1. Lines 266-267: “A variable number of Reed-Sternberg-like cells are EBV+ and express the EBV oncogenic protein the latent membrane protein-1 (LMP1) cells”.

The sentence has been corrected in the revised manuscript

  1. Table 1. An additional column is needed for related references

In agreement with the reviewer, we have included four references on the title of the Table:

IARC. Monographs on the Evaluation of Carcinogenic Risks to Humans. A review of human carcinogens. Part B: biological agents. IARC Working Group on the Evaluation of Carcinogenic Risks to Humans. Vol. 100, Lyon: 2012.

Carbone A, Borok M, Damania B, Gloghini A, Polizzotto MN, Jayanthan RK, Fajgenbaum DC, Bower M. Castleman disease. Nat Rev Dis Primers. 2021 Nov 25;7(1):84. doi: 10.1038/s41572-021-00317-7. PMID: 34824298; PMCID: PMC9584164.

Carbone A, Vaccher E, Gloghini A. Hematologic cancers in individuals infected by HIV. Blood. 2022 Feb 17;139(7):995-1012. doi: 10.1182/blood.2020005469. PMID: 34469512.

Carbone A, Chadburn A, Gloghini A, Vaccher E, Bower M. Immune deficiency/dysregulation -associated lymphoproliferative disorders. Revised classification and management. Blood Rev. 2024 Mar;64:101167. doi: 10.1016/j.blre.2023.101167. Epub 2024 Jan 5. PMID: 38195294

  1. Include a table summarizing treatment options by lymphoma subtype, including regimens, outcomes, and key trials.

As suggested by the Reviewer, we have included the new Table 2 that summarizes front-line treatment (Key trials) by lymphoma subtype, including regimens and outcomes.       

Minor issues

  • Line 59: “HIV my also directly contribute...” → “HIV may also directly contribute...”.
  • Line 126: “EBNA6 expressed by EBV-infected B cells may may lead...” → redundant “may”: “may lead”.
  • Lines 222-229. PI3K-Akt…. PI3K-Protein Kinase B.
  • Line 273: “based on identical morphologic, immunophenotypic and genotypic criteria as for lymphomas...” → Suggest: “based on the same morphologic, immunophenotypic, and genotypic criteria used in lymphomas...”.
  • Line 299: “use of cART increases this risk to about 20–30 times...” → Reference [86] does not clearly support this dramatic increase. Needs verification or rephrasing.
  • Section 3.5. Diffuse large B cell lymphoma (DLBCL). Since “DLBCL is the most common form of HIV-associated lymphoma”, it should be listed first.
  • Subtitles 5 Immunotherapeutic perspectives for lymphomas in PLWH; 5. Conclusions

All the minor issues have been corrected according to the Reviewer’s suggestions.

Round 2

Reviewer 2 Report

Comments and Suggestions for Authors
  1. The structure is still a problem in Sections 2 and 3. For Section 2, it is better to add subtitles highlighting several "indirect mechanisms"; For Section 3, remove the subtitle "Lymphoproliferative disorders of varied malignant potential", and explain the use of "polymorphic lymphoproliferative disorders", instead of "lymphoproliferative disorders". What are the differences? Sub-section "Lymphomas arising in PLWH" should be moved to "Introduction".
  2. Section 2.1. EBV and KSHV infections are deemed as "indirect mechanisms". This is a big question---In ARLs, is HIV infection the dominant cause, or is EBV or KSHV infection the dominant cause? Please clarify.
  3. "They (ARLs) are broadly categorized into Hodgkin lymphomas (HL) and Non-Hodgkin lymphomas (NHL)." This statement is misleading since lymphomas not associated with HIV are also categorized into HL and NHL.

Author Response

Response point by point to Reviewer 2 (Round 2)

  1. The structure is still a problem in Sections 2 and 3. For Section 2, it is better to add subtitles highlighting several "indirect mechanisms"; For Section 3, remove the subtitle "Lymphoproliferative disorders of varied malignant potential", and explain the use of "polymorphic lymphoproliferative disorders", instead of "lymphoproliferative disorders". What are the differences? Sub-section "Lymphomas arising in PLWH" should be moved to "Introduction".

In agreement with Reviewer’s suggestions, subtitles highlighting several “indirect” mechanisms have been included in Section 2 and the subtitle "Lymphoproliferative disorders of varied malignant potential" removed from Section 3. We have modified the initial part of the Section 3 starting with the description of the polymorphic lymphoproliferative disorders that encompass a broad spectrum of malignant phenotypes, therefore deserving a distinction from the truly malignant lymphomas (HL and NHL). The term polymorphic is used to highlight this distinctive feature of these disorders arising in different types of immunodeficiencies as compared to the generally more monomorphous characteristics of malignant lymphomas.

  1. Section 2.1. EBV and KSHV infections are deemed as "indirect mechanisms". This is a big question---In ARLs, is HIV infection the dominant cause, or is EBV or KSHV infection the dominant cause? Please clarify.

In PLWH, EBV and KSHV are the DIRECT driver of lymphomagenesis through the expression and function of the oncogenic proteins they encode. We agree with the reviewer that HIV has a critical role in promoting the oncogenic function of these viruses mainly by inducing immunosuppression, but HIV does not DIRECTLY infect the target B lymphocytes. Oncogenic viruses DIRECTLY infect the target cells and drive from within their transformation towards overt lymphomas. We have modified a sentence in the revised text to clarify this concept. “HIV-induced immunosuppression undermines immune surveillance, enabling the unrestricted proliferation of virus-infected cells and the aberrant expression of oncogenic viral proteins that directly drive lymphomagenesis”

  1. "They (ARLs) are broadly categorized into Hodgkin lymphomas (HL) and Non-Hodgkin lymphomas (NHL)." This statement is misleading since lymphomas not associated with HIV are also categorized into HL and NHL.

We have clarified this by modifying the following sentence: “HIV-associated malignant lymphomas are classified using the same morphologic, immunophenotypic, and genotypic criteria applied to lymphomas in immunocompetent individuals [5,88,89] and, similarly, are broadly categorized into Hodgkin lymphomas (HL) and Non-Hodgkin lymphomas (NHL).”